

# Sensitivity to the visual field origin of natural image patches in human low-level visual cortex

Damien J. Mannion

School of Psychology, UNSW Australia, Australia

## ABSTRACT

Asymmetries in the response to visual patterns in the upper and lower visual fields (above and below the centre of gaze) have been associated with ecological factors relating to the structure of typical visual environments. Here, we investigated whether the content of the upper and lower visual field representations in low-level regions of human visual cortex are specialised for visual patterns that arise from the upper and lower visual fields in natural images. We presented image patches, drawn from above or below the centre of gaze of an observer navigating a natural environment, to either the upper or lower visual fields of human participants ($n = 7$) while we used functional magnetic resonance imaging (fMRI) to measure the magnitude of evoked activity in the visual areas V1, V2, and V3. We found a significant interaction between the presentation location (upper or lower visual field) and the image patch source location (above or below fixation); the responses to lower visual field presentation were significantly greater for image patches sourced from below than above fixation, while the responses in the upper visual field were not significantly different for image patches sourced from above and below fixation. This finding demonstrates an association between the representation of the lower visual field in human visual cortex and the structure of the visual input that is likely to be encountered below the centre of gaze.

## INTRODUCTION

There are numerous reports of asymmetries in the visual system's representation of the upper and lower visual fields (that is, above and below an imaginary horizontal line that passes through the centre of gaze). Behavioural asymmetries between the upper and lower visual fields have been reported for a variety of visual tasks and stimuli (see *Previc, 1990*; *Danckert & Goodale, 2003*; *Karim & Kojima, 2010*, for reviews). For example, differences between the upper and lower visual fields have been reported for binocular rivalry (*Chen & He, 2003*), pointing at targets (*Danckert & Goodale, 2001*), and crowding (*He, Cavanagh & Intriligator, 1996*). The consequences of visual field loss are also asymmetric, with lower visual field loss having a greater functional impact than upper visual field loss (*Black, Wood & Lovie-Kitchin, 2011a*; *Black, Wood & Lovie-Kitchin, 2011b*). Neurally, the representations of the upper and lower visual fields in the low-level areas of human visual cortex have

Corresponding author
Damien J. Mannion,
d.mannion@unsw.edu.au

asymmetries in response (*Liu, Heeger & Carrasco, 2006*; *Chen, Yao & Liu, 2004*; *Hagler, 2014*) and architecture (*Eickhoff et al., 2008*).

Such behavioural and neural asymmetries between the upper and lower visual fields are often attributed to ecological factors relating to the structure of the visual environment (for example: *Skrandies, 1987*; *Tootell et al., 1988*; *Previc, 1990*; *Eickhoff et al., 2008*). The origins of the upper and lower visual field stimulation in outdoor environments are typically "sky" and "land," respectively (*Gibson, 1950*). These distal components can cause asymmetries in the statistics of the image patterns received by the upper and lower visual fields of the visual system (*Torralba & Oliva, 2003*; *Zanker & Zeil, 2005*; *Calow & Lappe, 2007*; *Schumann et al., 2008*).

However, there is little evidence that directly connects upper and lower visual field asymmetries in the visual system to the structure of natural images. We reasoned that, if such an association exists, the content of the upper and lower visual field representations in the low-level regions of human visual cortex would be specialised for visual patterns that arise from the upper and lower visual fields in natural images. Hence, the response in the representations of the upper and lower visual fields to a given natural image pattern would depend on its source in the visual field.

Here, we tested this hypothesis using functional magnetic resonance imaging (fMRI) of human low-level visual cortex. We used image patches obtained from above and below the centre of gaze of an observer freely-navigating an outdoor environment (*Schumann et al., 2008*), and presented them to the upper or lower visual fields of observers. We predicted that the amplitude of the blood-oxygen-level dependent (BOLD) signal in the low-level visual areas (V1, V2, and V3) would show an interaction between the presentation location (upper or lower visual field) and source location (above or below fixation); responses would be greater for image patches sourced from above than below fixation when presented in the upper visual field, and vice-versa for the lower visual field.

## MATERIALS & METHODS

### Participants

Seven participants, each with normal or corrected-to-normal vision, participated in the current study. Each participant gave their informed written consent and the study was approved by the Human Research Ethics Advisory Panel in the School of Psychology, UNSW Australia (2267/143–146).

### Apparatus

Functional imaging was conducted using a Philips 3T scanner. Images were collected with a $T_2^*$ sensitive gradient echo imaging pulse sequence (TR = 2 s, TE = 32 ms, flip angle = 90°, matrix = 96 × 96, FOV = 192 × 192 mm, voxel size = 2 mm isotropic) in 35 slices (interleaved, untilted, and coronal) covering the occipital lobes. The raw data are available at http://dx.doi.org/10.6080/K0JS9NC2.

Stimuli were displayed on an LCD monitor (BOLDscreen; Cambridge Research Systems, Kent, UK) with a spatial resolution of 1,920 × 1,200 pixels, temporal resolution

of 60 Hz, and mean luminance of 385 cd/m$^2$. The monitor output was linearized via correction of luminance values measured with a ColorCAL MKII colorimeter (Cambridge Research Systems, Kent, UK). The screen was viewed at a distance of 121.5 cm through a mirror mounted on the head coil, giving a viewing angle of 24.4° × 15.3°. Behavioural responses were indicated via a LU400-PAIR response pad (Cedrus Corporation, San Pedro, California, USA). Stimuli were displayed using PsychoPy 1.80.04 (*Peirce, 2007*). As detailed below, analyses were performed using FreeSurfer 5.3.0 (*Dale, Fischl & Sereno, 1999*; *Fischl, Sereno & Dale, 1999*), FSL 5.0.5 (*Smith et al., 2004*), and AFNI/SUMA (2014/06/06; *Cox, 1996*; *Saad et al., 2004*). Analysis code is available at https://bitbucket.org/djmannion/ul_sens_fmri_analysis.

## Stimuli

Each stimulus consisted of a set of apertures that were 4.5° in diameter and centred at 5.37° eccentricity in the four visual field quadrants (see Fig. 1). This aperture placement was designed to allow both unambiguous presentation in either the upper or lower visual field (hence, away from the horizontal meridian) and for the activation to be attributed to different visual areas with greater certainty (hence, away from the vertical meridian). On a given trial, the apertures in either the upper or lower visual field displayed natural image patches. The natural image patches were sourced from recordings by *Schumann et al. (2008)*, which are aligned to the centre of the gaze of an observer freely-navigating an environment (that is, the centre of the recorded image corresponded to the centre of the observer's gaze). From a set of 1,000 images from the "urban" environment recordings (sampled randomly from the free navigation segments of the videos, subject to the constraint that a sampled image was not acquired at the time of a fast eye movement), we randomly chose 30 images to comprise the set for this study. For each image in this set, we extracted square patches (128 pixels in length) from the centre of the four image quadrants. During presentation, such patches were windowed by a circular mask with a raised cosine edge transition and upsampled (by a factor of 2.75) via linear interpolation. The patches were presented on a background of mean luminance that was overlaid with several isopolar and isoeccentric lines to promote stable fixation (*Hansen, Kay & Gallant, 2007*; *Schira, Wade & Tyler, 2007*), and lines were also continuously present to mark the boundaries of the circular patch masks. Example stimuli are shown in Fig. 1, and the stimuli for all images, source locations, and presentation locations are shown in Fig. S1.

## Design

The experiment used a condition-rich rapid event-related design protocol (*Kriegeskorte, Mur & Bandettini, 2008*). Each scanning run consisted of two sets of 83 events—one event sequence for presentation in the upper visual field and one event sequence for presentation in the lower visual field. In each set of 83 events, there were 15 null events in which no stimulus was displayed and 60 events in which a pair of image patches was displayed within the relevant apertures. The 60 image events consisted of a single presentation of image patches sourced from either above or below fixation in the 30 images used in the study (see Stimuli). On a given image event, both apertures (left and right) displayed content from

**Source**: below, **Presentation**: upper

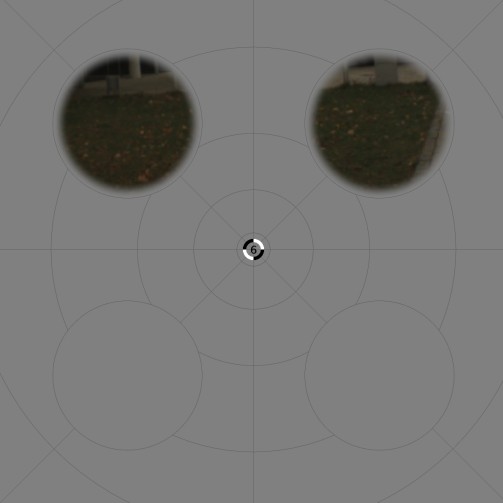

**Source**: above, **Presentation**: upper

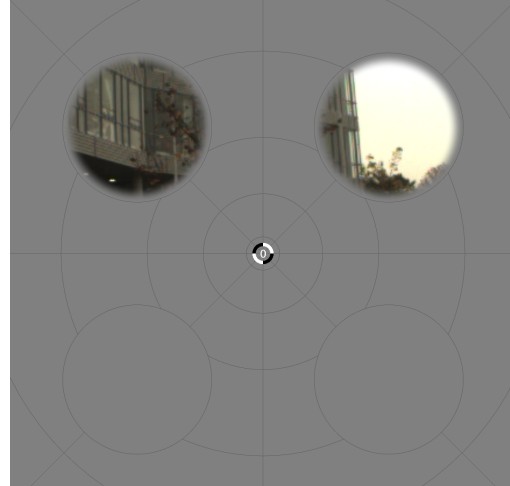

**Source**: below, **Presentation**: lower

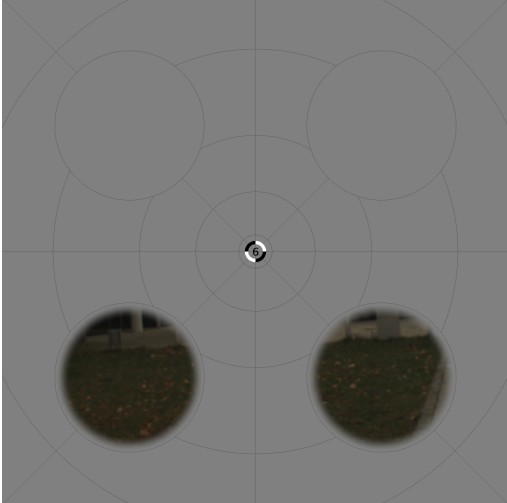

**Source**: above, **Presentation**: lower

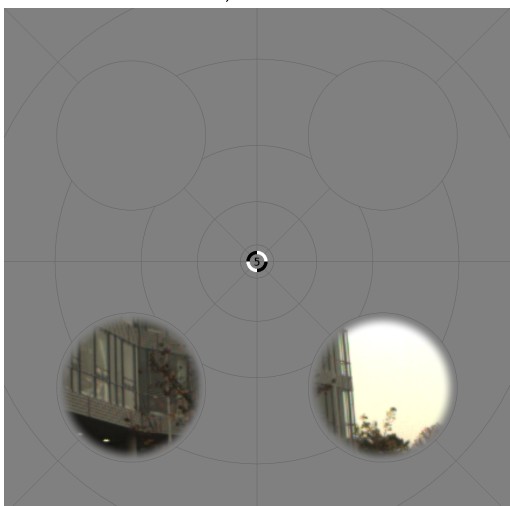

**Figure 1 Stimulus and experimental design.** Each stimulus consisted of apertures in the visual field quadrants. On a given trial, the apertures in either the upper or lower visual field (*presentation location*) displayed a pair of natural image patches. These patches were sourced from either above or below the centre of gaze of an observer navigating a natural environment (*source location*). Each panel shows these four experimental conditions for an example natural image. The number shown at central fixation relates to a behavioural task; see Design for details.

the same image. The ordering of the null and image events was randomised, and the final 8 events were replicated and prepended to the sequence to allow an initial period of each run to be removed while maintaining balanced trial counts.

There was a 4 s inter-event interval, and, for image events, the image patches were presented for the initial 1 s of the event with a raised-cosine contrast ramp in the initial and final 0.1 s. To prevent perceptual grouping or contextual interactions between the above and below fixation presentations, the timing sequence for either the upper or lower visual field events (chosen randomly for each run) was advanced by 2 s. Hence, image patches were

never simultaneously visible in the upper and lower visual fields. Overall, each run was 332 s in duration and participants completed 10 runs that were collected in a single session.

Participants were engaged in a challenging behavioural task during each run (see *Mannion, Kersten & Olman, 2013*, for details). Briefly, participants were required to respond with a button press whenever one of two target digits and polarities (black or white) was presented in a rapidly updated (3 Hz) stream at central fixation (see Fig. 1). This task had no direct relevance to the aims of this study, but was designed to divert participant's attention away from the stimuli and hence to limit the effects of potentially unequal attentional allocation to different stimulus conditions.

### Anatomical acquisition and processing

A $T_1$-weighted anatomical image (sagittal MP-RAGE, 1 mm isotropic resolution) was collected from each participant in a separate session. FreeSurfer (*Dale, Fischl & Sereno, 1999*; *Fischl, Sereno & Dale, 1999*) was used for segmentation, cortical surface reconstruction, and surface inflation and flattening of each participant's anatomical image.

### Visual area definition

The V1, V2, and V3 visual areas were defined based on analysis of functional acquisitions, obtained in a separate scanning session, that followed standard procedures for the delineation of retinotopic regions in human visual cortex; see *Mannion, Kersten & Olman (2013)* for details. Briefly, participants observed four runs of a clockwise/anti-clockwise rotating wedge stimulus and two runs of an expanding/contracting ring stimulus (*DeYoe et al., 1996*; *Engel, Glover & Wandell, 1997*; *Hansen, Kay & Gallant, 2007*; *Larsson & Heeger, 2006*; *Sereno et al., 1995*; *Schira, Wade & Tyler, 2007*), and the data was analysed via phase-encoding methods (*Engel, 2012*) to establish visual field preferences over the cortical surface. The angular and eccentricity phase maps were used to manually define each participant's V1, V2, and V3.

### Pre-processing

Functional images were corrected for differences in slice acquisition timing using AFNI. Motion correction was performed using AFNI; motion estimates were obtained both for each individual run and across runs before being combined and used to resample the functional images using windowed sinc interpolation. The participant's anatomical image was then coregistered with a mean of all the functional images using AFNI's `3dAllineate`, using a local Pearson correlation cost function (*Saad et al., 2009*) and six free parameters (three translation and three rotation). Coarse registration parameters were determined manually and passed to the registration routine to provide initial estimates and to constrain the range of reasonable transformation parameter values. The motion-corrected functional data were then projected onto the cortical surface by averaging between the white and pial boundaries (identified with FreeSurfer) using AFNI/SUMA. No specific spatial smoothing was applied. All analyses were performed on the nodes of this surface domain representation in the participant's native brain space.

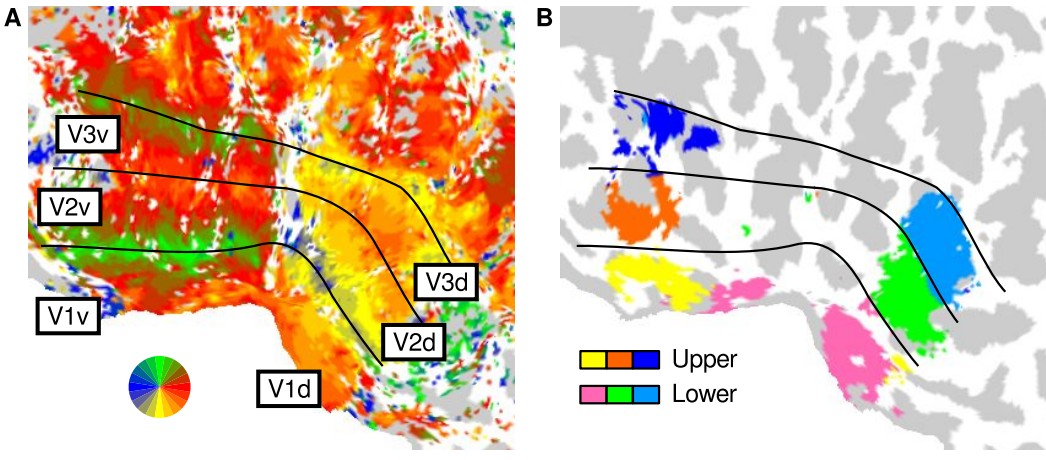

**Figure 2 Cortical regions corresponding to upper and lower visual field presentation.** The panels show the flattened posterior left hemisphere of an example participant. (A) Mapping of the preferred angular location in the visual field, as identified using a standard rotating wedge paradigm. The colours correspond to the map of the visual field, and the lines demarcate the low-level visual areas. (B) Localised regions of interest for the upper and lower visual field presentation in the current study.

## Analysis

The response timeseries for each participant were analysed within a general linear model (GLM) framework using AFNI, separately for the upper and lower visual fields. The event onset sequences were convolved with SPM's canonical haemodynamic response function and included as regressors in the GLM design matrix. The first 16 volumes (32 s) of each run were censored in the analysis, leaving 1,500 data timepoints (150 per run for 10 runs). Polynomials up to the third degree were included as regressors for each run to capture low temporal frequency intensity fluctuations. The GLM was estimated via AFNI's `3dREMLfit`, which accounts for noise temporal correlations via a voxelwise ARMA(1,1) model.

### *Localiser*

We first executed a GLM analysis (see Analysis) to localise regions of each visual area and presentation position (upper or lower visual field) that were responsive to the presence of image structure. Accordingly, we used a single regressor to model the response to image patches sourced from the lower and upper visual fields. This produced a pair of $t$ values for each node on the cortical surface; one $t$ value for the upper visual field presentation sequence and one $t$ value for lower. Such $t$ maps were thresholded at $p < 0.01$ to create a mask of upper-responsive and lower-responsive regions within each visual area. These upper and lower presentation masks were largely confined to ventral and dorsal subregions, respectively—as expected given the retinotopic organisation of V1, V2, and V3 (*Schira et al., 2009*)—and are shown for an example participant in Fig. 2.

### Response amplitude

To analyse the response amplitude elicited by image patches sourced from above and below fixation and presented in either the upper or lower visual field, we first spatially averaged the response timecourses for the upper and lower visual field presentation masks (see Localiser for how such masks were defined and Fig. 2 for example masks). We then used such data to conduct separate GLMs for the upper and lower visual field presentation conditions, with each GLM using separate regressors for image patches sourced from the lower and upper visual fields. The beta weights obtained from each GLM for each condition were then converted to percent signal change via division by the temporal average of the polynomial regressor timecourse.

This procedure yielded an estimate of the response, in percent signal change units, elicited by each image (30), patch source (above, below), presentation location (upper, lower), visual area (V1, V2, V3), and participant. We performed a three-way repeated measures ANOVA (visual area: V1, V2, V3; presentation: upper, lower; source: above, below) on such response amplitude estimates (averaged over images). Violations of the assumption of sphericity were corrected by using Huynh-Feldt coefficients to reduce the effective degrees of freedom when assessing statistical significance, where appropriate. The complete results from this analysis are presented in Table S1.

## Image patch analysis

For each image (30), vertical source location (above, below), and horizontal source location (left, right), we extracted the pixel values from a square region (353 × 353) in the display window as viewed by observers in the experiment. Each pixel was then transformed into DKL space (*Derrington, Krauskopf & Lennie, 1984*) using the methods described by *Lu & Dosher (2014)* and *Brainard (1996)*. Briefly, the CIE coordinates of the red, green, and blue channels of the monitor (as measured with a colorimeter) were used to transform the pixel values into DKL space: an achromatic axis and two isoluminant chromatic axes.

We characterised each image, vertical source location, horizontal source location, and DKL axis by the mean and standard deviation of its pixel values and by the output of Gabor filters with differing spatial frequency and orientation preferences. The filter bank consisted of Gabor functions with spatial frequencies of 1, 2, 4, 8, and 16 cycles per degree of visual angle and orientations of 0°, 45°, 90°, and 135°. The sinusoid carrier in each Gabor had a phase of 90° (odd-symmetric), and the standard deviation of the Gaussian envelope was set so the Gabor had a spatial frequency bandwidth of one octave. For each image patch, the response at each Gabor spatial frequency and orientation was taken as the square root of the sum, over space, of the squared output of the convolution of the image patch with the Gabor filter. We averaged over the different orientations when evaluating the spatial frequency content, and vice-versa when evaluating the orientation content, of the image patches.

## RESULTS AND DISCUSSION

We find a significant interaction between the presentation and source locations ($F_{1,6} = 11.19, p = 0.016$), as is apparent in Fig. 3. Because there was little evidence to indicate that

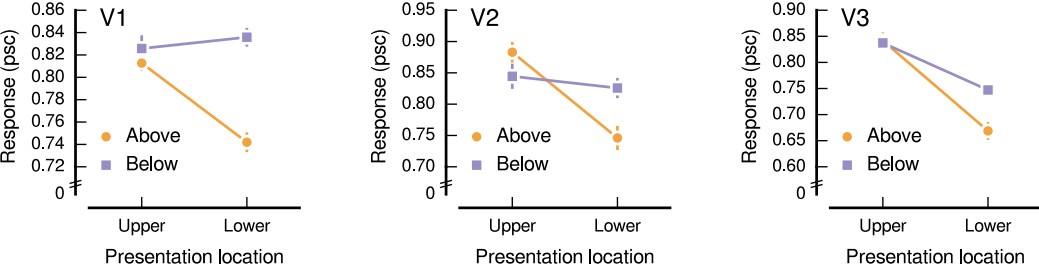

**Figure 3 Response amplitudes to natural image patches.** The horizontal axis shows the presentation location, which was either the upper or the lower visual field. The vertical axis shows the response amplitude in units of percent signal change (psc), averaged over participants (normalized for differences in overall activation). Lines show the response to image patches sourced from above (orange, circle) and below (purple, square) fixation. Panels show the low-level visual areas V1, V2, and V3. Error bars show standard error of the normalized responses.

this interaction depended on the visual area ($F_{1.9, 11.4} = 1.52, p = 0.260$), we combined across the visual areas (V1, V2, and V3) for subsequent analyses. Responses to images sourced from below fixation were higher than those from above fixation when presented in the lower visual field, with the below and above sources eliciting an average response amplitude, in units of percent signal change (psc), of 0.80 psc (SE = 0.02) and 0.72 psc (SE = 0.02), respectively. This difference was statistically significant ($t_6 = 5.17, p = 0.002$). When presented in the upper visual field, images sourced from below fixation elicited comparable response amplitudes to those from above fixation (below: mean = 0.84 psc, SE = 0.03; above: mean = 0.85 psc, SE = 0.02; $t_6 = 0.38, p = 0.719$).

Hence, our hypothesis was partially supported; the response in low-level human visual cortex to natural image patches depended on whether the patches were presented in the same visual field location from which they were sourced—but only for presentation in the lower visual field and not in the upper visual field. The possibility that the lower visual field is specialised for particular stimulus patterns and functions while the upper visual field is largely invariant has been noted by *Young (1990)*. Indeed, the majority of reported asymmetries between the upper and lower visual fields have favoured the lower visual field (*Karim & Kojima, 2010*). However, as most previous studies have investigated the difference between upper and lower visual field stimulation rather than the interaction with stimulus properties targeted to potential upper and lower visual field specialisations, it is difficult to confidently assess the existence of complementary specialisations, or lack thereof, in the upper and lower visual fields. The results obtained here, however, are suggestive of a specialisation that is predominantly limited to the lower visual field.

As discussed, the responses to image patches sourced from above and below fixation were unequal when presented in the lower visual field—those sourced from above fixation evoked a reduced response. To gain insight into the characteristics of the above and below fixation image patches that may underpin this effect, we conducted an exploratory analysis into the response evoked by each individual image patch pair (left and right). We computed the difference between the responses evoked in the upper and lower visual fields by each image patch pair, ranked the image patch pairs on such differences, and identified the top

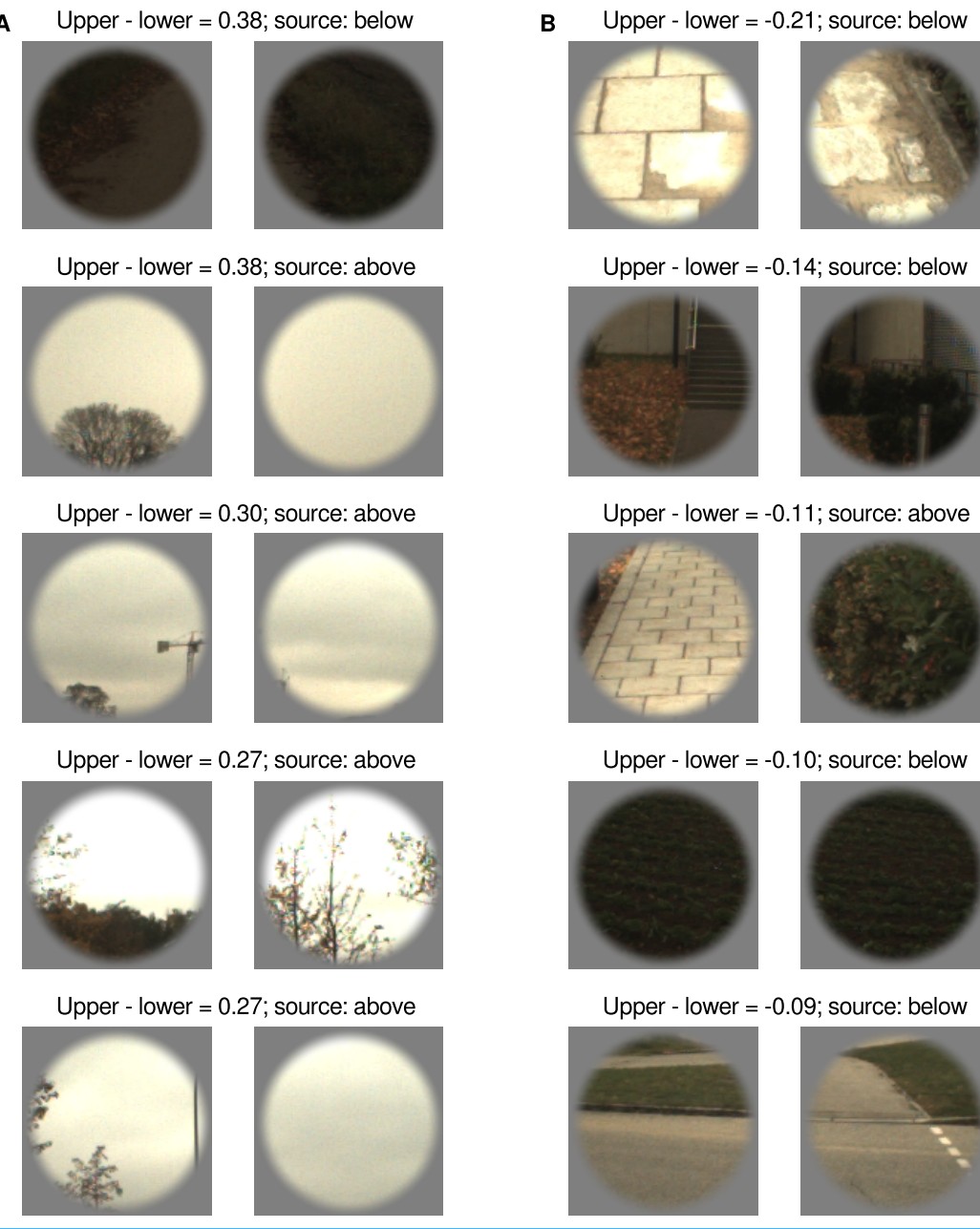

**Figure 4 Image patch pairs evoking the largest differences between upper and lower visual field presentation.** Columns (A) and (B) show the top five upper-preferred and lower-preferred image patch pairs, respectively. The difference between upper and lower visual field presentation is given in percent signal change units (psc), averaged over participants. The image patch pair 'source' refers to whether it came from above or below the fixation of an observer navigating a natural environment.

five 'upper-preferred' and 'lower-preferred' image patch pairs. We find that four out of the top five patches for both upper-preferred and lower-preferred are from the matching source location (above fixation for upper-preferred, below fixation for lower-preferred). As evident in Fig. 4, there are clear qualitative differences between those image pairings that

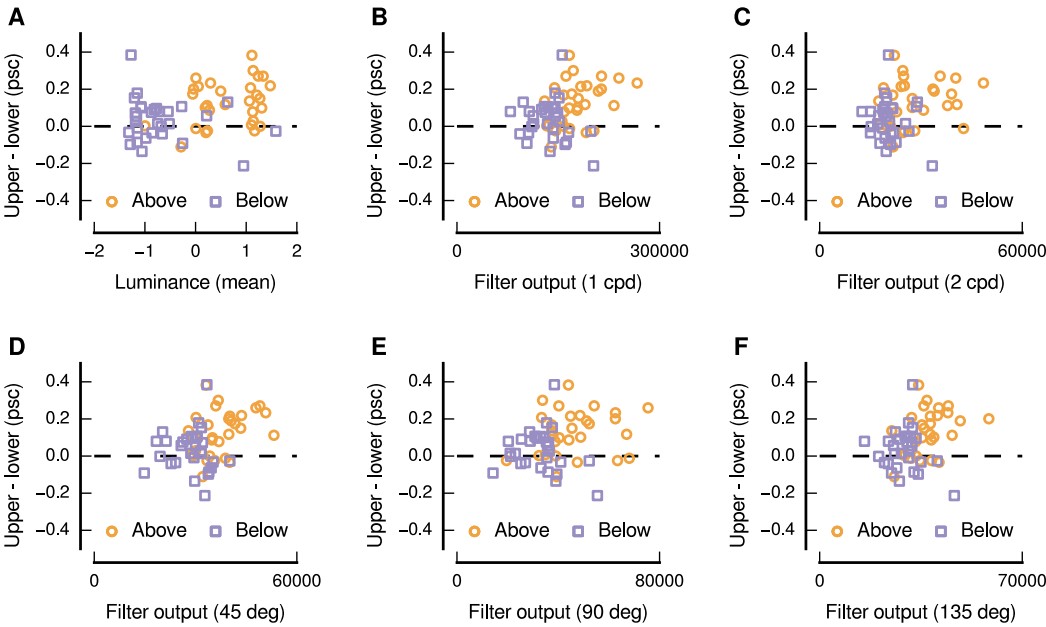

**Figure 5 Relationship between image characteristics and the difference in the magnitude of activity evoked in low-level visual cortex for upper and lower visual field presentations.** Each panel shows an image characteristic for which there was a significant positive monotonic relationship, and all panels depict image characteristics based on the luminance channel in DKL space. In each panel, each point represents a particular image and source location (above: orange circles, below: purple squares).

evoke a greater response in the upper visual field and those that evoke a greater response in the lower visual field. The image patch pairs preferred by the upper visual field consist of regions of largely uniform luminance (either bright or dark), whereas those preferred by the lower visual field are more variegated. There is also an apparent distal dissociation, with the upper-preferred image patch pairs mostly depicting the sky and the lower-preferred image patch pairs mostly depicting the ground.

We also performed a quantitative analysis of the relationship between the upper and lower visual field response differences and the characteristics of the image patches sourced from above and below the centre of gaze. Consistent with the above qualitative impression, we find that there were significant positive monotonic relationships between the upper and lower visual field differences and several image characteristics (as shown in Fig. 5). Considering the distributions of pixel values, we find that higher levels of mean luminance were related to greater differences in the responses evoked by upper and lower visual field presentation (Spearman's $r = 0.31, p = 0.016$). This positive relationship is ostensibly in opposition to the depiction in Fig. 4, in which dark patches are shown to have evoked a high level of upper preference—however Fig. 5A indicates that the response to this pair is not indicative of the general relationship. No significant relationship was evident for the mean values in the chromatic channels, or for the standard deviation of the pixel intensities for any of the channels.

In addition to such analyses, based on distributions of pixel values, we also used a set of Gabor filters (*Daugman, 1985*; *Gabor, 1946*; *Marčelja, 1980*) to simulate the response

of spatial frequency and orientation-selective neurons in low-level visual cortex. We find that higher levels of output for filters with peak spatial frequency preferences of 1 and 2 cycles per degree of visual angle, applied to the luminance channel of the image patches, were also related to greater differences in responses evoked by upper and lower visual field presentation (Spearman's $r = 0.32, p = 0.013$; Spearman's $r = 0.26, p = 0.049$; for spatial frequencies of 1 and 2 cycles per degree of visual angle, respectively), as shown in Figs. 5B and 5C. We also find that higher levels of output for filters with peak orientation preferences of 45° (right-tilted), 90° (vertical), and 135° (left-tilted), applied to the luminance channel of the image patches, were also related to greater differences in responses evoked by upper and lower visual field presentation (Spearman's $r = 0.34, p = 0.008$; Spearman's $r = 0.25, p = 0.050$; Spearman's $r = 0.39, p = 0.002$; for orientations of 45°, 90°, and 135°, respectively), as shown in Figs. 5D, 5E, and 5F. No significant relationships were evident for any other spatial frequencies or orientations in the luminance channel, or for any spatial frequency or orientation in the chromatic channels.

The preceding exploratory analysis suggests a potential limitation of the current study. To present image patches from a particular source (above or below fixation) in the opposite location (lower or upper visual field), we simply translated the patches either up or down. This approach means that the image patches are not mirror symmetric about the horizontal meridian in the upper and lower visual field presentation conditions—that is, they are not vertically flipped in addition to being translated. We adopted this approach because it is more compatible with conventional movement of the head and eyes rather than movement that would cause vertical flipping (such as viewing from between one's legs, as described by *Von Helmholtz (1925)*). However, it does lead to the current experiment being prone to the confounding effects of the radial bias (*Sasaki et al., 2006*; *Mannion, McDonald & Clifford, 2010*), which is a preference for edges that are collinear with the centre of gaze. The upper-preferred and lower-preferred image patches shown in Fig. 4 do not have an abundance of radially-dominated structure, suggesting that it is unlikely to explain the effects observed here. This conclusion is supported by an analysis of the orientation structure of the image patches—there were no significant differences between the output of Gabor filters oriented at 45° and 135°, which would be expected if a reliable radial bias was present in the image patches.

The current study also has further limitations that are suggestive of avenues for future research. First, we are unable to generalise the results beyond the particular set of images that were used. Although the images were chosen from an environment that is reasonably ecologically representative, in that it contained a mixture of 'natural' and man-made structure, it is unclear if the asymmetries reported here would hold across a variety of environments. Second, we cannot precisely identify image characteristics that underlie the apparent asymmetry between the upper and lower visual fields. However, the qualitative and quantitative image analyses are suggestive of potential characteristics that may be further explored and manipulated in future studies. Finally, we did not monitor participant eye movements and thus cannot completely rule out a contribution from differential eye movements to the observed results. However, there are no apparent differences in

performance on the behavioural task (Fig. S2) or in the quality of the GLM model after event onset (Fig. S3) across conditions—consistent with the expectation that observers were compliant with the instructions to direct their gaze to the central fixation task throughout the experiment.

## CONCLUSION

We found that the response in human low-level visual cortex to natural image patches presented in the lower visual field depended on whether the patches were sourced from above or below the gaze of an observer navigating a natural environment—patches sourced from below fixation evoked significantly greater activity than patches sourced from above fixation, when presented in the lower visual field. This finding demonstrates that asymmetries in the content of the upper and lower visual field representations can be associated with the visual patterns that are likely to be encountered across the visual field in natural environments. Ultimately, we hope that this work will motivate future investigations towards the goal of understanding how visual field representations relate to the structure and function of the input that arises from above and below fixation in natural vision.

## ACKNOWLEDGEMENTS

We thank P. König and colleagues for sharing the video recordings from which the stimulus images were obtained, Matthew Patten for scanning assistance, Colin Clifford for sharing equipment, and the radiographers at St. Vincents Hospital for scanner operation. We also thank the Collaborative Research in Computational Neuroscience project (http://crcns.org) for hosting the raw data.

### Funding

This work was supported by an Early Career Research grant from the Faculty of Science, UNSW Australia. The funders had no role in study design, data collection and analysis, decision to publish, or preparation of the manuscript.

### Grant Disclosures

The following grant information was disclosed by the author:
Early Career Research grant from the Faculty of Science, UNSW Australia.

### Competing Interests

The author declares there is no competing interests.

### Author Contributions

- Damien J. Mannion conceived and designed the experiments, performed the experiments, analyzed the data, contributed reagents/materials/analysis tools, wrote the paper, prepared figures and/or tables, reviewed drafts of the paper.

## Human Ethics

The following information was supplied relating to ethical approvals (i.e., approving body and any reference numbers):

UNSW Australia Human Research Ethics Advisory Panel (Psychology), approval #2267/143-146.

## Supplemental Information

Supplemental information for this article can be found online at http://dx.doi.org/10.7717/peerj.1038#supplemental-information.

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
