# Peer review of "Sensitivity to the visual field origin of natural image patches in human low-level visual cortex"

_PeerJ, doi:10.7717/peerj.1038_

## Round 0.1 · original submission · Major Revisions

The reviewers comments are very clear and reasonable. Please address them thoroughly. Reviewer 2's comments, in particular, concerning additional controls, strike me as very helpful to this line of research. Having said that, both reviewers agree that your specific conclusions stand up with the current design. So if you decide to not do the control experiments proscribed by R2, to add to this manuscript, and instead simply address their lack thereof as a potential weakness of the current design, that will be acceptable. But I do hope you do those experiments here (and if not, in a future study, as R2 is correct that they would be informative).

·

Basic reporting

No Comments

Experimental design

There is a central task, as mentioned in the caption of Fig 1. But the task was not described in the paper.

Validity of the findings

It was not clear whether "presentation location" in Fig 2 is the same as cortical location--I had assumed the upper location corresponded to ventral visual cortex, and lower location corresponded to dorsal visual cortex. Is this the case? It was never explicitly stated, except tin the "Localiser" section of the Method, something to that nature was mentioned. "These upper and lower presentation masks were largely confined to ventral and dorsal subregions, respectively". Some clarification would be good. I'd suggest to show some example data from the localiser, perhaps on a cortical surface, to inform the reader regarding from what regions of brain the fMRI response was obtained.

Additional comments

The experiment seemed to be well-conducted and results are quite interesting. My other comment is whether the author can show some image statistics for the stimulus set. It would be useful to know how the images from above vs. below sources differ on basic dimensions such as luminance, contrast, orientation, spatial frequency, and color etc. I think knowing this would help us understand better why there is a differential neural response to these stimuli, and can potentially inform future studies to examine the cause of such difference.

Reviewer 2 ·

Basic reporting

This manuscript describes a novel property of fMRI bold responses in early visual cortex (V1-V3). Using images that fall in the upper or lower visual fields during typical viewing, they show that fMRI responses are stronger when these images are then presented at the correct location, compared to when they are presented in the other field. This is mainly a property of the lower field representation, which responds less strongly to images that typically fall on the upper field.

The work is clearly described and technically sound.

Experimental design

The experiment has very limited scope, showing the result of just one experiment and no controls. So we are left with almost no idea about what is responsible for this effect. Might it be as simple as differences in mean luminance (which might then mean that the state of photoreceptor adaptation is different in the two fields)? This might even conceivably give rise to differences in pupil size. Does the effect survive normalizing the contrast? Although these possibilities do not change the conclusion that something is different in the processing of upper and lower visual fields, it makes it impossible to say anything about how interesting this adaptation is.
It would have been very helpful to also include Left/Right swapping of images, as this would help to show that this was really specific to the upper/lower distinction. It would also have dealt with the concern about the effect possibly reflected a bias for radial orientations that was in fact the same in upper and lower fields, which at present they cannot refute (but at least it is acknowledged).
Although subjects had a demanding task at fixation, they did not measure eye movements. Any tendency to move the eyes in the incongruous stimulus condition might give rise to their main effect.

Validity of the findings

The result is robust and statistically sound. I do not doubt the validity of the result itself. but it admits of many interpretations. Ideally i would like to seem some further experiments investigating the possible causes.

---

## Round 0.2 · Minor Revisions

Please address Reviewer #3's remaining minor revision requests.

·

Basic reporting

I noted there is a requirement to share raw data, and that reviewer is supposed to comment on that. I can't tell from the manuscript whether the data are shared. I assume the journal will enforce that.

Experimental design

No comments.

Validity of the findings

No comments.

Additional comments

The author has addressed my concerns satisfactorily.

Reviewer 2 ·

Basic reporting

Everything is clearly described.

Experimental design

Design is satisfactory as far as it goes, but makes little attempt to understand possible reasons for the result.

Validity of the findings

Data are straightforward enough, but have many possible explanations.

Additional comments

The authors have responded to all of the queries. In several places the responses take the form of accepting the limitations of the study, so I am left with the feeling that this is a limited study. They observe a difference between congruent and incongruent situations. There may be a trivial an uninteresting explanation for this, they don’t know. That’s a “follow up question”. But the manuscript lays all of this out pretty clearly, so its an editorial decision whether reports of this type are of interest.
The analysis they present in the rebuttal of performance on the demanding task does not by any means exclude possible variations in eye position. I think they should add a sentence in the discussion noting that they did not measure eye position so there is a possibility that changes in eye position behavior might explain the result also. And they can then give reasons they think this unlikely.

---

## Round 0.3 · accepted · Accept

You will need to post your raw data as required by journal policy. Open a free account on crcns.org and follow the guidelines to post your data at your earliest convenience.